# Cell wall dynamics stabilize tip growth in a filamentous fungus

**Louis Chevalier[1,2], Mario Pinar[3], Rémi Le Borgne[1], Catherine Durieu[1], Miguel A. Peñalva[3], Arezki Boudaoud[4]\*, Nicolas Minc[1,2]\***

**1** Université Paris Cité, CNRS, Institut Jacques Monod, Paris, France, **2** Equipe Labellisée LIGUE Contre le Cancer, Paris, France, **3** Department of Cellular And Molecular Biology, Centro de Investigaciones Biológicas Margarita Salas, Madrid, Spain, **4** LadHyX, CNRS, Ecole polytechnique, Institut Polytechnique de Paris, Palaiseau, France

\* arezki.boudaoud@polytechnique.edu (AB); nicolas.minc@ijm.fr (NM)

**Data Availability Statement:** "All relevant data are within the paper and its Supporting Information files (S1 Data)."

**Funding:** This work was supported by grants from the "Fondation de la Recherche Médicale" (n°

## Abstract

Hyphal tip growth allows filamentous fungi to colonize space, reproduce, or infect. It features remarkable morphogenetic plasticity including unusually fast elongation rates, tip turning, branching, or bulging. These shape changes are all driven from the expansion of a protective cell wall (CW) secreted from apical pools of exocytic vesicles. How CW secretion, remodeling, and deformation are modulated in concert to support rapid tip growth and morphogenesis while ensuring surface integrity remains poorly understood. We implemented subresolution imaging to map the dynamics of CW thickness and secretory vesicles in *Aspergillus nidulans*. We found that tip growth is associated with balanced rates of CW secretion and expansion, which limit temporal fluctuations in CW thickness, elongation speed, and vesicle amount, to less than 10% to 20%. Affecting this balance through modulations of growth or trafficking yield to near-immediate changes in CW thickness, mechanics, and shape. We developed a model with mechanical feedback that accounts for steady states of hyphal growth as well as rapid adaptation of CW mechanics and vesicle recruitment to different perturbations. These data provide unprecedented details on how CW dynamics emerges from material secretion and expansion, to stabilize fungal tip growth as well as promote its morphogenetic plasticity.

## Introduction

Filamentous fungi are generally nonmotile but exploit fast polar tip growth for surface colonization, mating, or host infection [1]. In typical vegetative life cycles, for instance, fungal spores germinate to outgrow polarized hyphae that expand rapidly at their tips and undergo branching, turning, and sometimes fusion to generate the complex mycelium network [2,3]. Hyphal cell shape and growth are defined by the dynamic expansion of their cell wall (CW), which surrounds and protects the plasma membrane [4,5]. In general, however, how the CW undergoes such rapid and diverse shape changes while ensuring surface mechanical integrity remains poorly understood.

Fungal CWs are composed of reticulated polysaccharides including chitin, α- and β-glucan and mannose polymers, as well as remodeling enzymes like hydrolases and transferases [6]. Post-Golgi RAB11 exocytic vesicles (EVs) are thought to secrete a subset of sugars and proteins

13171) to L.C., the "Spain's Ministerio de Ciencia e Innovación" (grant RTI2018-093344-B100) and the "Comunidad de Madrid and he European Regional Development and European Social Funds" (grant S2017/BMD-3691) to M.A.P, the "La Ligue Contre le Cancer" (EL2021.LNCC/ NiM) and the "European Research Council" (ERC CoG "Forcaster" no. 647073) to N.M., as well as the "Agence Nationale pour la Recherche" (ANR, "CellWallSense" no. ANR-20-CE13-0003-02) to N.M. and A.B. "The funders had no role in study design, data collection and analysis, decision to publish, or preparation of the manuscript."

**Competing interests:** "The authors have declared that no competing interests exist."

**Abbreviations:** ConA, Concanavalin A; CW, cell wall; EM, electron microscopy; EV, exocytic vesicle; FWMH, full width at mid height; HPF, high pressure freezing; PDMS, polydimethylsiloxane; PH, Pleckstrin homology; PM, plasma membrane; WGA, wheat germ agglutinin; WMM, watch minimal medium; WT, wild-type.

into the CW and also to carry transmembrane enzymes to the plasma membrane that catalyze the elongation of other sets of sugars [3,7]. Therefore, secretory vesicles may promote both CW material assembly and extensibility needed to support mechanical stability and growth [4,8]. Vesicles are trafficked toward the hyphal tip along microtubules tracks and recycled via a subapical endocytic ring domain [3,9,10]. At cell tips, they are clustered by F-actin and myosin type V motors around a dense reservoir called the Spitzenkörper, thought to be adapted to rapid hyphal growth in many but not all fungal species [11–18]. Secretory vesicles radiate from this local reservoir, through transport and diffusion, to eventually tether and fuse with the plasma membrane and fuel CW assembly [17–21]. Chemical or genetic conditions that affect the polarized trafficking of EVs halt tip growth and often yield to defects in tip shape [22–24]. Accordingly, variations in EVs concentration, apical domain sizes and shapes have been correlated to tip expansion speeds, and diameters in multiple fungi [25–27]. Yet, to date, a detailed assessment of how secretory vesicle pools contribute to actual CW material assembly and expansion in live growing cells is still lacking.

The composition and assembly of the fungal CW define its material properties that underpin its ability to protect hyphal cells and allow them to grow. CWs have thicknesses that may vary between approximately 50 to 500 nm and bulk elastic moduli of approximately 10 to 100 of MPa, akin to a material like rubber [28,29]. The CW is put under tension by a large cytoplasmic pressure of several atmospheres, called turgor, which is osmotically generated. Turgor serves as a core mechanical engine to deform freshly assembled CW portions at cell tips and, thus, power cell growth, but also entails risk of CW failure and cell death [8]. CW growth for tip elongation has been modeled in multiple instances. Some models primarily focused on secretory vesicle supplies disregarding contributions from turgor and CW material properties [30–32]. Others have been based on frameworks of visco-elasto-plastic thin shells, assuming that newly assembled CW portions at the apex undergo plastic irreversible deformation above a threshold stress but used simplified descriptions of material supply from secretory vesicles [33–38]. Interestingly, both modeling and experimental work have suggested the existence of mechanical feedbacks, whereby enhanced strain rates in the CW may promote the recruitment or stability of polar secretory domains [34,39–41]. Despite the potential predictive power of these models, quantitative comparison with experimental data has been limited. Accordingly, we still lack quantitative models and experiments that enable to understand how the dynamics of CW secretion, expansion, and mechanics may be regulated during fungal growth and shape changes.

Here, we used *Aspergillus nidulans*, an established tractable model fungus that features rapid hyphal growth [42,43], to understand in quantitative terms how CWs are built and remodeled from EV pools during tip growth. We developed a super-resolution live imaging method to map CW thickness around growing hyphal cells, which we combine with the dynamic imaging of RAB11-labeled EVs, and with biochemical and genetic interventions affecting growth, turgor, trafficking, or secretion. We propose a mathematical model for tip growth that we systematically test and calibrate against dynamic perturbations. Combined with experimental findings, this model suggests the existence of a mechanical feedback from CW growth to vesicle accumulation that accounts for stable steady-state hyphal growth at various elongation speeds.

## Results

### Monitoring cell wall thickness in live fungal hyphae

To understand how the CW is dynamically assembled and deformed during rapid fungal hyphal growth, we adapted a previous subresolution imaging method to monitor CW thickness in live yeast cells, to *Aspergillus nidulans* [28,44]. We labeled the plasma membrane that lines the inner face of the CW with a Pleckstrin homology (PH) domain fused to GFP [45]. To

label the outer face of the CW, we added in the medium, either Concanavalin A (ConA), a lectin that binds α-D-mannosyl and α-D-glucosyl sugar residues, or wheat germ agglutinin (WGA), which binds N-acetyl-D-glucosamine residues, tagged with Alexa-647 fluorophores. These lectins, predicted to be larger than the typical CW pore size, preferentially decorate surface exposed polysaccharides, as evidenced from the lack of staining in internal septa (Figs 1A, 1B and S1A) [46,47]. We acquired two-color confocal images and extracted the distance between the two fluorescent peaks across the cell surface by fitting them with Gaussian functions. This analysis was repeated all around cells, using automated homemade scripts that segment the whole cell contour, register color spatial shifts, and compensate for peak asymmetries associated to different signal-to-noise ratio inside and outside cells [28,48] (Fig 1C). Using this method, we could map CW thickness on long contours of up to a hundred of micrometers in live hyphal cells, with a precision that we estimated to be approximately 10 to 20 nm and a spatial resolution along the cell surface of approximately 500 nm (Fig 1D and 1E). Although adding lectins did not grossly affect hyphal morphology and growth, the method was often limited spatially by out-of-focus parts of the hyphae in the field of view, and temporally by phototoxic effects that tended to affect growth above approximately 30 consecutive fluorescent images (S1B Fig).

In mature hyphae growing at the bottom of dishes in liquid watch minimal medium (WMM), this live method yielded a mean CW thickness of 78 +/− 17 nm, ($n$ = 53 cells, +/− indicate +/− SD) when labeling the CW with WGA, and 65 +/− 21 nm ($n$ = 89 cells) with ConA (S1C Fig). These values were in similar range when using other membrane dyes or proteins (S1D Fig). Growing hyphae in rich MCA medium or between agar pads covered with a dialysis membrane and a coverslip, both led to higher thicknesses, of 83 +/− 21 nm and 90 +/− 14 nm, respectively, and to wider cells, plausibly reflecting adaptation of the CW to environmental conditions (S1E and S1F Fig). Importantly, these values were comparable to those obtained by electron microscopy using either chemical fixation (83 ± 10 nm, $n$ = 8) or high pressure freezing (HPF) (82 ± 22 nm, $n$ = 8) of cells grown in liquid-rich medium, and also aligned with previous reports [29,49] (Fig 1F and 1G). In addition, measurements of CW thickness of germinated spores confirmed that they have a much thicker CW than hyphae (137 +− 26 nm, $n$ = 24), similar to previous reported values (Figs 1E, 1H and S1G) [50]. Finally, using this method, we could film cells and map spatiotemporal variations in CW thickness along both germling tubes that outgrew from spores, or more mature hyphae, over periods of approximately 20 to 30 min during which cells expanded approximately 10 to 20 μm in length (Fig 1H and S1 Movie). Therefore, it was possible to monitor CW thickness dynamics in rapidly growing hyphae of a filamentous fungus.

## A spatial gradient of cell wall stiffness is associated to hyphal polar growth

Many models for tip growth posit that cell tips shall feature softer and/or thinner CWs to account for polarized CW deformation [51,52]. In mature hyphae, we found that CW thickness exhibited a relatively shallow gradient, with tips being on average only approximately 13% thinner than cell sides. In addition, we noted that a fraction of cells exhibited a reversed pattern, with a thicker CW at cell tips (Fig 2A). Furthermore, inspections of time-lapse sequences taken at one frame per minute suggested that this CW thickness polarity could even become inverted in a time course as short as approximately few minutes, indicative of rapid CW remodeling activity at cell tips (S1 Movie). We conclude that CW thickness gradients may not be sufficient to polarize CW mechanics for tip growth in these cells.

To compute local values of CW elasticity around cells, we built on the analysis of thickness, h, to compute the CW Young's modulus, Y, which reflects bulk material properties and its

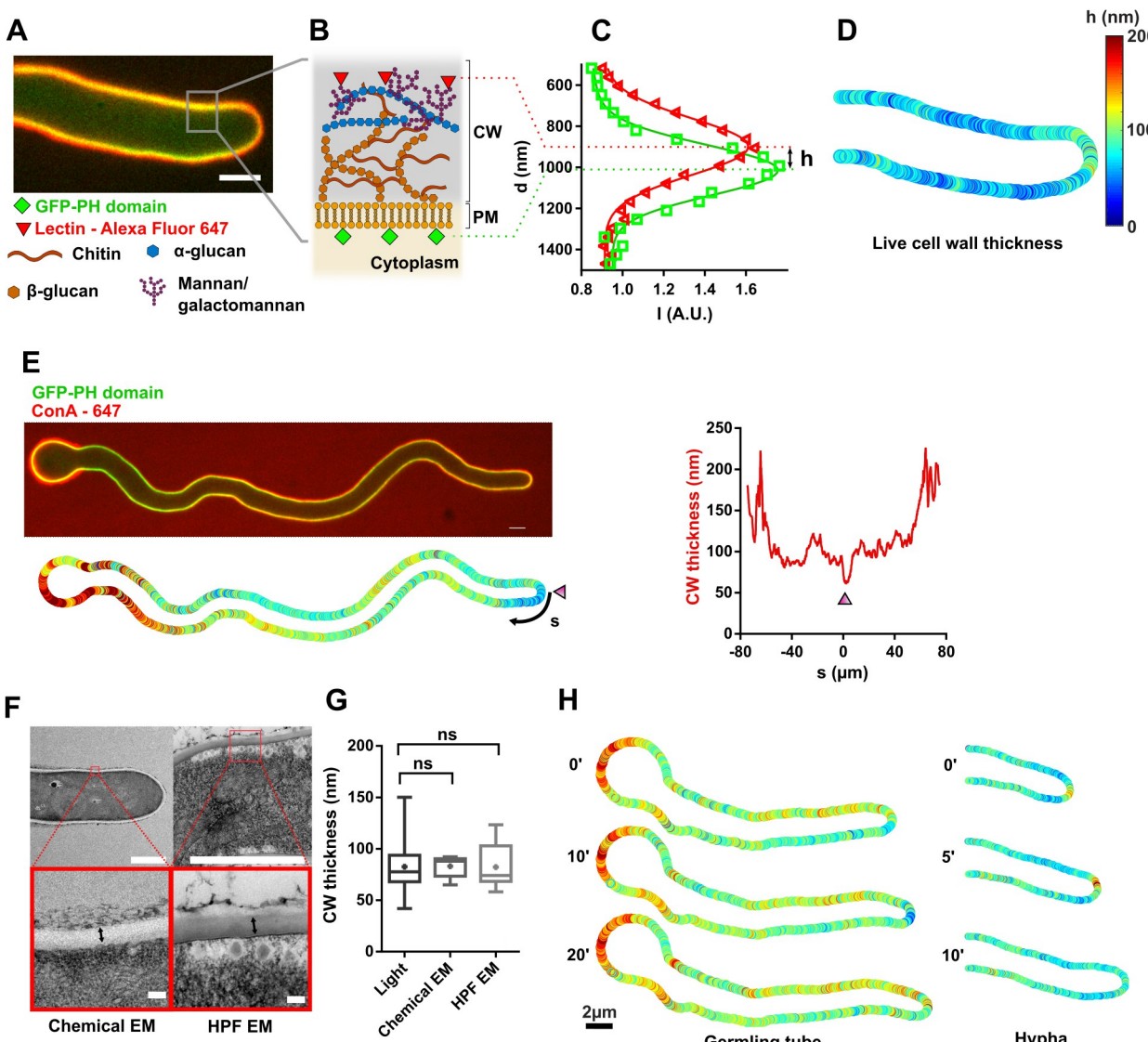

**Fig 1. A subresolution imaging method to monitor CW thickness dynamics in live fungal hyphae.** (**A**) Mid-slice confocal image of a live *Aspergillus nidulans* hyphal cell expressing a GFP-PH domain (plasma membrane, CW inner surface) and labeled with the lectin ConA-Alexafluor 647 (CW outer surface). (**B**) Scheme of CW organization showing different polysaccharides and positions of the two fluorescent signals. (**C**) Gaussian fits of the signal distribution of each fluorophores across the CW. The distance between the two Gaussians peaks allows to compute a local value of CW thickness, h. (**D**) Resulting CW thickness color map around the live cell presented in panel A. (**E**) Top: Mid-slice confocal image of a germling tube, with the spore body visible on the left. Bottom: corresponding CW thickness color map. Right: Measured CW thickness profile along the cell plotted as a function of the arclength distance, s, with s = 0 being the center of the cell tip (marked with an arrowhead). (**F**) Measurement of CW thickness, marked with a double arrow, in electron microscopy, using chemical fixation (left) or HPF (right). (**G**) CW thicknesses measured using our live-microscopy method (*n* = 81 cells) and electron microscopy from chemically fixed (*n* = 8) or high pressure–frozen cells (*n* = 8). (**H**) Time lapse of CW thickness maps, in relatively slow elongating germling tube and a faster mature hypha. Scale bars: (A, E, H): 2 μm. (F) top: 2 μm, bottom: 100 nm. Error bars correspond to +/− SD. Results were compared by using a two-tailed Mann–Whitney test. n.s, *P* > 0.05. The data underlying the graphs can be found in S1 Data. CW, cell wall; EM, electron microscopy; HPF, high pressure freezing; PM, plasma membrane.

surface modulus, σ, which is the product of thickness and Young's modulus, hY, and represents the apparent CW stiffness [28]. We imaged cells to map CW thickness and rapidly photoablated the CW using a focalized UV laser [34]. This caused the pressurized cytosolic material to flow out of cells within seconds, yielding cell deflation and CW elastic relaxation. CW

relaxation allowed to compute a local elastic strain, $R_1^* = \frac{R_1 - R_0}{R_0}$, with $R_0$ and $R_1$ the local cell radii before and after deflation. This first showed that the CW relaxed twice as much along the radial axis as compared to the longitudinal axis of the cell, suggesting relatively low anisotropies in the CW material (S2A Fig) [33]. Second, it allowed to compute local values of Y/P, with P the turgor pressure, from force balance relationships in the pressurized CW, with $\frac{Y_{tip}}{P} = \frac{R_{t1}}{2h_{tip}R_{t1}^*}$ at cell tips, and $\frac{Y_{side}}{P} = \frac{R_1}{h_{side}R_1^*}$ for lateral CWs (Figs 2B and S2B) [28,53].

Therefore, in order to compute exact local values of CW elastic moduli, we measured turgor pressure. We assumed turgor to be homogenous within hyphal compartments and monitored shape changes of the CW, as above, but in response to hyperosmotic shocks of different magnitudes. This led to estimates of turgor values of P ~ 1.1 to 1.3 MPa, from the medium osmolarity needed to shrink cells as much as with CW photoablation [53,54] (Figs 2C and S2C). Together, these analyses show that the CW Young's modulus follows a steep gradient from $Y_{tip}$ = 64 ± 45 MPa at cell tips, up to $Y_{side}$ = 210 ± 103 MPa on cell sides approximately 10 to 14 μm away from cell tips (Fig 2E) [29]. Combining thickness and Young's modulus, we obtained local values of CW surface moduli or apparent stiffness (hY) that evolved from 4.3 ± 2.5 N/m at cell tips up to 15.3 ± 6.2 N/m on cell sides (Fig 2F). Such gradients in CW stiffness might reflect spatial differences in the cross-linking of CW components. Therefore, hyphal polar growth in *Aspergillus nidulans* is accompanied by a steep gradient of surface stiffness, dominated by

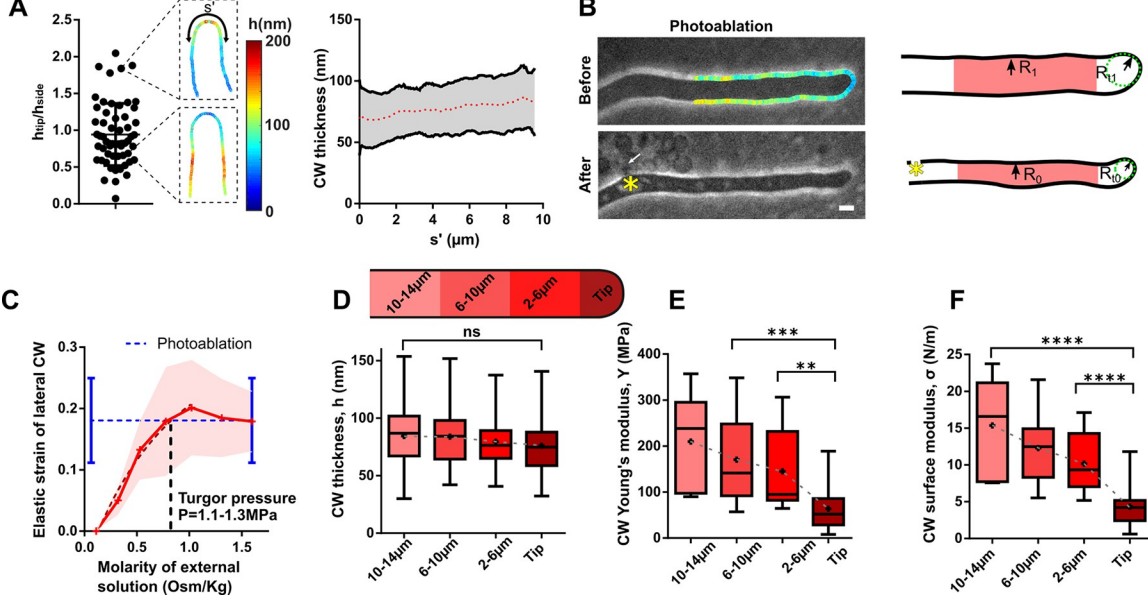

**Fig 2. Spatial gradients of CW elasticity along fungal hyphae. (A)** Left: Distribution of the ratio of the CW thickness at cell tips to that on cell sides, with two exemplary thickness color maps of cells with different thickness polarity. Right: CW thickness gradient along the cell contour, using a symmetrized arclength distance, s', as coordinate (s' = 0 being the tip) (*n* = 58 cells). **(B)** Method used to compute local CW Young's modulus around hyphal cells. Left: Bright-field images of the same cell, before (top) and after (bottom) photoablation, with measured CW thickness map before ablation. The asterisk marks the site of photoablation, and the arrow points at cytoplasmic material leaking out of the cell. Right: Segmented CW boundaries of the same cell before and after ablation used to compute the local elastic strain and deduce local values of CW Young's elastic modulus, from values of thickness and elastic strains. **(C)** Elastic strain of the lateral CW measured as the relative radial shrinkage $\frac{R_1 - R_0}{R_0}$ for osmotic shocks of different magnitudes, and compared with the value obtained from CW photoablation assays (blue dotted lines) (*n* > 13 cells for each osmolyte concentration). The intersection of the two curves provides an estimate of the external molarity needed to reduce turgor to zero, and thus an estimate of turgor pressure. **(D-F)** Distribution of CW thickness, h, Young's modulus, Y, and surface modulus, σ = hY along the hyphae, as defined in the scheme (*n* > 38 cells for the CW thickness, *n* > 7 cells for Y and σ). Scale bar, 2 μm. Error bars correspond to +/− SD. Results were compared by using a two-tailed Mann–Whitney test. n.s, P > 0.05; **, P < 0.01, ***, P < 0.001, ****, P < 0.0001. The data underlying the graphs can be found in S1 Data. CW, cell wall.

spatial variations in bulk material properties, with tips CWs being approximately 2 to 3× softer than lateral CWs.

## Spatial patterns of secretory vesicle accumulation and cell wall mechanics

To assay if these local variations of CW thickness and mechanics at cell tips reflected polarized CW synthesis, vesicle transport, or endocytosis, we used three-color imaging to coimage CW thickness with important regulators of these processes. This included mCherry-labeled type V myosin motor, MyoV-mCherry, which functions to transport EVs and marks the Spitzenkör-per; mCherry-RAB11 to directly visualize the pool of post-Golgi RAB11 EVs; the transmembrane chitin synthase, mCherry-ChsB, which serves as a proxy for CW synthesis; and Lifeact-tdTomato as reporter for F-actin [9,55,56]. As previously reported, Lifeact preferentially labeled endocytic patches along a subapical collar, while all other factors localized to cell tips [10,57] (Figs 3A–3C and S3A). To assay which of these markers may best represent local mechanical variations of tip CWs, we selected cells exhibiting a marked gradient in CW thickness being either thicker or thinner at cell tips and compared the width of the polarity zone formed by different markers with the width of the tip thickness profile. This analysis revealed that the MyoV-mCherry signal was more focused than zones of CW thickness variations, while mCherry-ChsB had a broader distribution. Similarly, the zones delimited by the F-actin endocytic ring were about twice as large as the width of the thickness gradient. Interestingly, mCherry-RAB11, provided the closest width to that of CW thickness gradients (Figs 3D, 3E, S3B and S3C). Accordingly, affecting the distribution of mCherry-RAB11, using a *myoVΔ* mutant, led to wider cells, with significantly wider distributions in both EV domains and CW thickness gradients at cell tips (Fig 3F and 3G) [15]. Thus, although these results do not rule out contributions from polarized transport, endocytosis, and CW synthesis to both EVs and CW thickness distribution, they suggest that the mCherry-RAB11 signal may be used as a close proxy for CW mechanical changes at cell tips. Together, these analyses directly highlight in living cells the spatial relationships between exocytic carrier distribution and local modulations in CW mechanics [33].

## A model for fungal tip growth, cell wall secretion, assembly, and remodeling

We next explored the temporal relationships between EVs accumulation at cell tips, CW thickness, and growth. This analysis rested on an optimization of tip growth assays, in which hyphal tips were kept in focus by placing them between agar pads covered with a dialysis membrane and coverslips, as well as an automated analysis of mCherry-RAB11 signal at cell tips [27] (Fig 4A and 4B and S2 Movie). Remarkably, in spite of the very fast elongation speeds of hyphal cells, we found that growth speeds, mCherry-RAB11 intensity, and CW thickness at cell tips exhibited limited temporal fluctuations with an amplitude of approximately 10% to 20% of their mean value (Fig 4C). Importantly, temporal cross-correlation analyses did not reveal any systematic relationship between a transient increase or decrease of one these parameters with those of another one (S4A Fig). This suggests that rapid tip growth is a relatively stable process, in which dynamic feedback systems maintain vesicle recruitment, CW assembly, and expansion around steady-state values.

To begin to understand how CW synthesis, remodeling, and mechanics are coupled to promote steady-state rapid tip growth in filamentous fungi, we developed a simple one-dimensional analytical model (Fig 5A). We considered a general form of mass conservation for the CW material with a source term associated with synthesis and a sink term to mechanical thinning inherent to CW expansion during tip growth. We neglected a putative contribution of

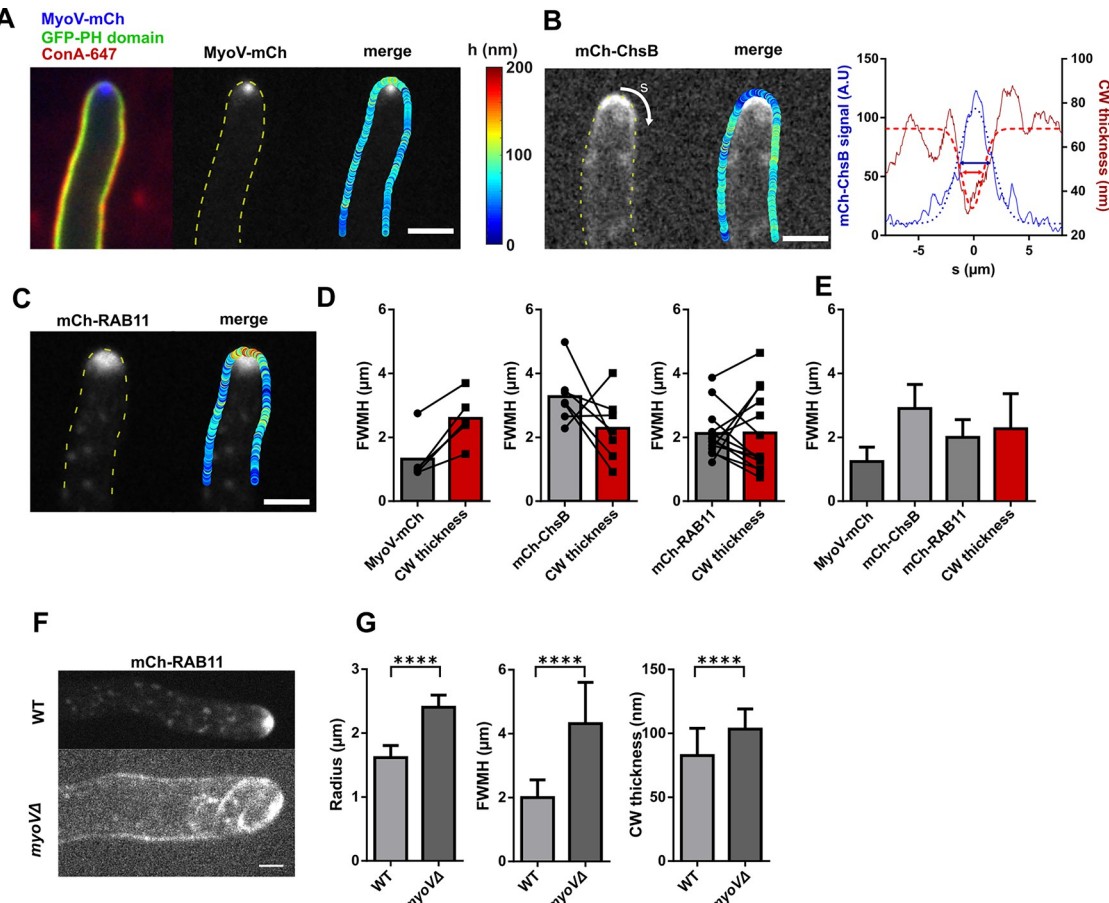

**Fig 3. Spatial distribution of downstream regulators of CW assembly and CW thickness profiles.** (**A-C**) Distribution of different tagged polar regulators of CW assembly, together with CW thickness profiles, for Myosin type V: MyoV-mCherry (**A**), the chitin synthase, mCherry-ChsB (**B**), and the post-Golgi vesicle labeling GTPase mCherry-RAB11 (**C**). In B, the profile of CW thickness and signal of mCherry-ChsB are plotted as a function of the arclength (s) and fitted with Gaussians to compute the FWMH, for both distributions (double arrows). (**D**) FWMH for each protein fluorescent signal and corresponding FWMH of the CW thickness profile, with individual cells connected by black lines. (**E**) FWMH distribution of different polar factors and CW thickness ($n$ = 16, 13, 29, and 25 cells). (**F**) Images of the EVs marker mCherry-RAB11 in a WT and in a *myoVΔ* mutant cell. (**G**) Hyphal radius, FWMH of mCherry-RAB11 and CW thickness profiles and mean values of tip CW thickness for WT and *myoVΔ* mutant ($n$ = 49, 30 cells). Scale bars, 2 μm. Error bars correspond to +/− SD. Results were compared by using a two-tailed Mann–Whitney test. ****, $P < 0.0001$. The data underlying the graphs can be found in S1 Data. CW, cell wall; FWMH, full width at mid height; WT, wild-type.

advection, whereby wall material could flow out of cell tips faster than cells elongate [38], as tracking fiducial CW portions with higher or lower thicknesses, did not reveal any notable differential speed between translated CW portions and tip growth (S4B Fig). This yields to an evolution equation of CW thickness, $h$:

$$\frac{dh}{dt} = \gamma EV - Gh, \tag{1}$$

with $EV$ denoting the concentration of RAB11 EVs at cell tips, $\gamma$ a constant, and $G$ the strain rate of tip CW, which equals the cell surface growth speed divided by the square of the tip radius of curvature, $R$. We assumed CW deformation to follow an elastoplastic behavior, with a strain rate proportional to the elastic strain, $PR/Yh$, in excess of a threshold plastic yield strain

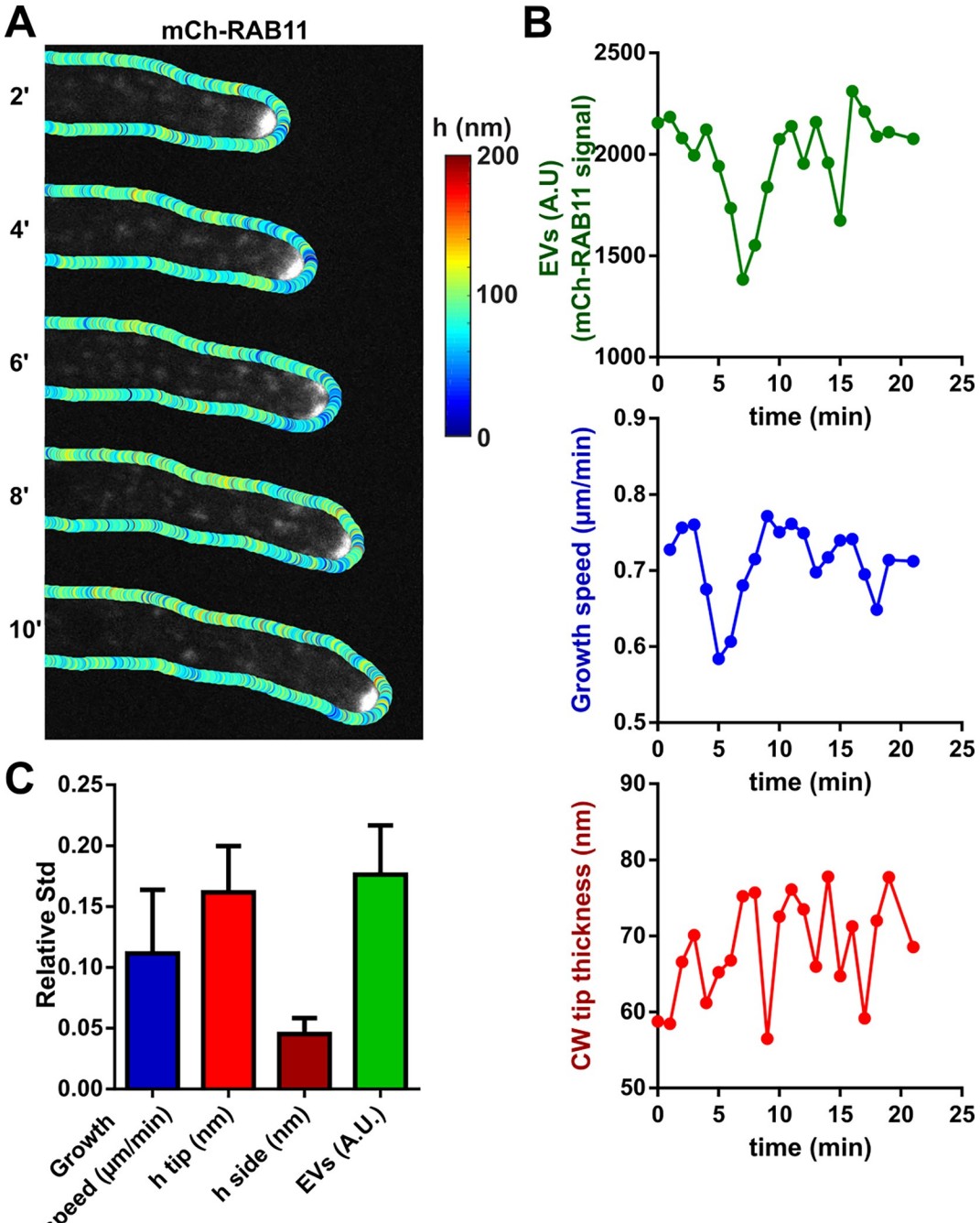

**Fig 4. Dynamic tracking of post-Golgi EVs together with growth and CW thickness.** (**A**) Time lapse of a growing WT hypha, with CW thickness maps overlaid on mCherry-RAB11 signal. (**B**) Quantification of the time evolution of the concentration of post-Golgi EVs at cell tips, tip elongation speed, and CW thickness. (**C**) Relative Stds computed over multiple time lapses for growth speed, CW thickness on cell sides and cell tips, and EVs intensity ($n = 10$ time lapses in different cells). Scale bar, 2 μm. Error bars correspond to +/− SD. The data underlying the graphs can be found in S1 Data. CW, cell wall; EV, exocytic vesicle; Std, standard deviation; WT, wild-type.

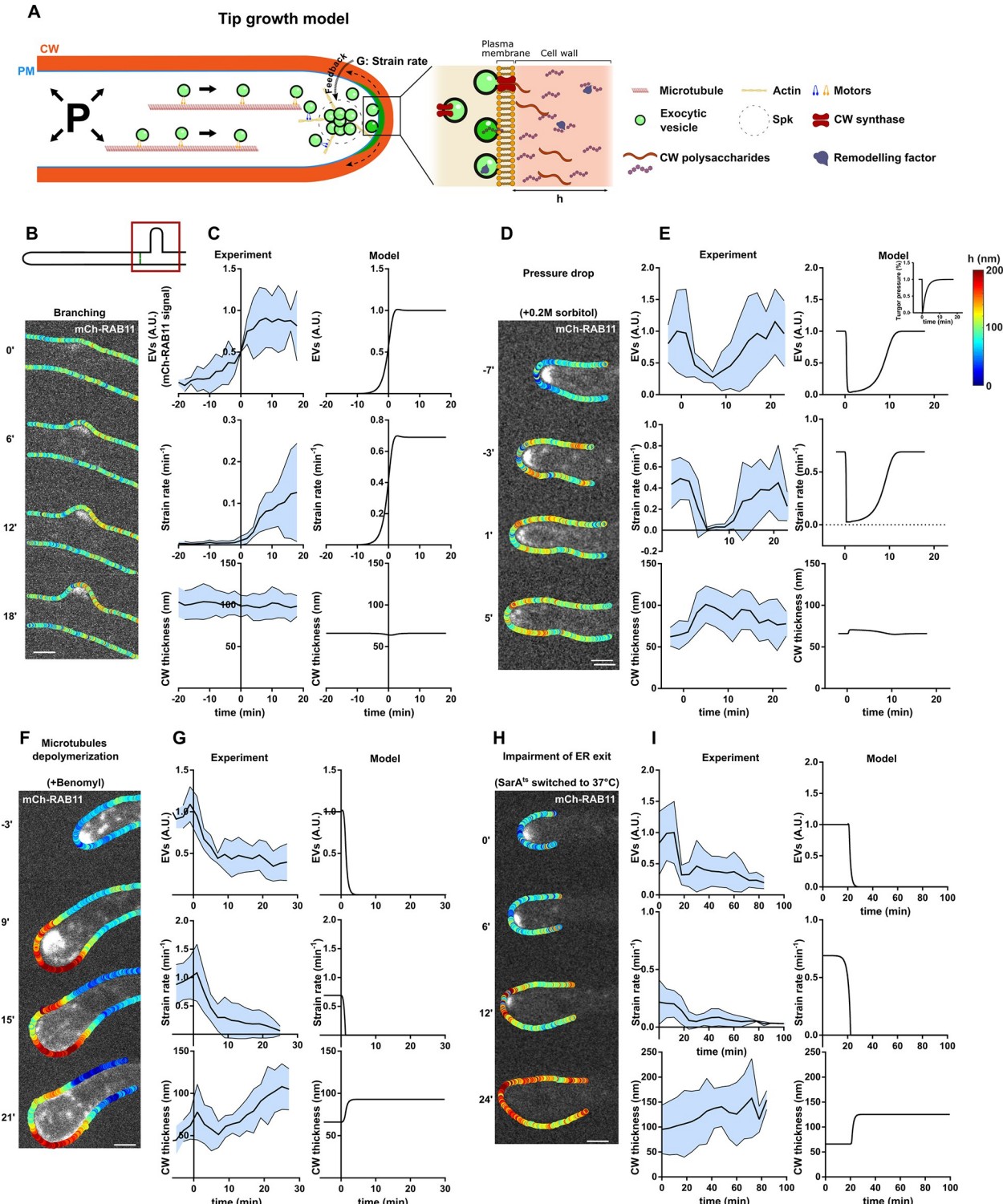

**Fig 5. Dynamic cotracking of EVs, CW thickness, and deformation during abrupt changes in cell growth or secretion.** (**A**) Scheme of a hyphal fungal tip, highlighting key assumptions of the mathematical model for tip growth: EVs cluster in the Spitzenkörper and radiate to promote the secretion of CW polysaccharides as well as remodeling enzymes. This allows the CW to thicken as well as deform under stresses created by turgor, at a strain rate, G. The pool of EVs is fed by polarized trafficking and other sources of recycling in proportion to the strain rate of the CW at cell tips (mechanical feedback). (**B**) Time lapse of EV accumulation (mCherry-RAB11) and CW thickness during a lateral branching event. (**C**) Dynamics of EVs concentration, CW strain rate, and thickness in branching cells (*n* = 7) and corresponding model outputs. The origin of time is defined as the first visible emergence of the new branch. (**D**) Time lapse of a growing cell rinsed at time t = 0 with medium supplemented with 0.2 M sorbitol to

reduce turgor pressure. (**E**) Dynamics of EVs concentration, CW strain rate, and thickness before and after the osmotic shock (*n* = 10) and corresponding model outputs. The inset in the top curve represents how turgor was modulated in the model to simulate the initial rapid drop and subsequent adaption. (**F**) Time lapse of a cell treated with benomyl to depolymerize microtubules, displaying CW thickness profiles overlaid with EVs. (**G**) Dynamics of EVs concentration, CW strain rate, and thickness before and after benomyl addition (*n* = 5) and corresponding model outputs. (**H**) Time lapse of a temperature-sensitive *sarA6 ts* mutant cell switched to restrictive temperature at time 0, showing the CW thickness profiles overlaid with EVs. (**I**) Dynamics of EVs concentration, CW strain rate, and thickness in *sarA6 ts* cells at restrictive temperature (*n* = 8) and corresponding model outputs. Scale bars, 2 μm. In experimental plots presented in 5C, 5E, 5G, and 5I, the black line represents the mean of the data and the blue shade the standard deviation. The data underlying the graphs can be found in S1 Data. CW, cell wall; EV, exocytic vesicle; PM, plasma membrane.

$\epsilon$, and to the concentration of CW remodeling factors, c, with a coefficient $\mu$ [35,38].

$$G = \mu c \left( \frac{PR}{Yh} - \epsilon \right) \tag{2}$$

To describe the dynamics of EVs concentration, we first posited that it decayed by feeding the CW with a transfer constant α. Second, we assumed that it was alimented in proportion to the strain rate, positing the existence of a mechanical feedback with a feedback parameter φ [39,40,44]. Importantly, stability analyses of the model showed that such feedback was strictly required to generate stable steady-state growth as observed in experiments (S4C Fig). Furthermore, other types of feedback based on direct stress or elastic strain sensing did not allow to generate stable hyphal growth, supporting the assumption of mechanical feedback from strain rates (S4D Fig). This leads to the dynamic of the EV concentration *EV*:

$$\frac{dEV}{dt} = \phi G - \alpha EV \tag{3}$$

While EVs bring CW synthases to the membrane and polysaccharides into the CW, to thicken it, they also secrete remodeling elements to promote CW extensibility. We thus added to the model an equation describing the dynamic of the surface concentration, *hc*, of these elements in the CW:

$$\frac{d(hc)}{dt} = \beta EV - Ghc \tag{4}$$

in which β in an incorporation coefficient. Eq 4 combined with Eqs 1 and 2 yields to the evolution of strain rates:

$$\frac{dG}{dt} = \eta \, EV \frac{1 - \theta h}{h^2} - \gamma \frac{2 - \theta h}{(1 - \theta h)h} G \, EV + \frac{1}{1 - \theta h} G^2 \tag{5}$$

with new parameters $\eta = \frac{\beta \mu PR}{Y}$ and $\theta = \frac{\epsilon Y}{PR}$ that are related to the previously defined parameters.

Eqs 1, 3, and 5 define a system of ordinary differential equations of order 1 for RAB11 EVs concentration level *EV*, the strain rate (growth), *G*, and tip CW thickness, *h*, with 5 adjustable positive parameters. Stability analysis of the model shows that it has only one stationary and stable point with nonzero values for G and *EV*, suggesting it may in principle capture the general process of hyphal tip growth. In order to make semiquantitative comparisons between the model and measured experimental values, we calibrated it using measured tip thickness and strain rate and pressure drops needed to stop growth in wild-type (WT) mature hyphae.

As our model is based upon a major assumption of a feedback system that positively couples vesicle concentration at cell tips with CW strain rates (or equivalently growth rates), we sought to provide direct experimental support for this assumption. Such hypothesis is based on the general observation made in fission yeast and *Candida albicans* that polarity domains

tend to detach from cell tips when growth is hindered by the presence of a mechanical barrier and reform in new axis away from the obstacle to promote growth along a mechanically favorable direction [34,40,41,51,58]. To test if polar domains of RAB11-labeled EVs in *Aspergillus nidulans* follow similar behavior, we grew hyphae against microfabricated PDMS elastic obstacles. We observed that when the contact between the hyphal tip and the obstacle was firm and maintained, the mCherry-RAB11 domain began to spread and decay in intensity, concomitant with a slow-down of tip growth and a slight flattening and bulging of the tip. In 79% of cases, EVs dispersal was followed by the progressive reformation of a single bright EVs domain that drove tip growth in a direction near orthogonal to the initial growth direction. In the remaining 21% cases, cells assembled two domains enriched in EVs, yielding to apical branching, with branches that grew in opposite directions (S5A–S5C Fig and S3 Movie). Importantly, such apical branching was never observed in our normal growth conditions with no obstacles. These observations suggest that a mechanical hindrance of tip growth can strongly affect EVs domain stability.

To assay if such behavior mirrored our hypothesis of mechanical feedback in the model, we modeled hyphal contact with the obstacle and subsequent escape by solving numerically the dynamical model, starting by a progressive and partial drop of turgor, followed by a restoration of turgor values. As expected, this allowed the model to account for the transient slowdown in tip elongation rate (strain rate). Importantly, given the hypothesis of feedback that couple growth and EVs accumulation, the model also reproduced the decay in EVs accumulation followed by a restoration to initial levels upon growth reorientation. Finally, the model predicted that such adaptation only had minor impact on CW thickness (S5B Fig). However, the presence of the obstacle prevented lectins to diffuse and label the CW surface at the contact site, hampering us to compute CW thickness experimentally in these assays. Together, these results support the general assumption of mechanical feedback in the model.

## Dynamic coevolution of CW thickness, tip growth and exocytosis, during hyphal tip shape changes

We next sought to test the model against abrupt changes in growth or secretion. One first natural instance during which growth or secretion may rapidly evolve in mycelial colonies is de novo tip growth at emerging lateral branches [59,60]. We monitored branch formation by focusing on hyphal compartments bound by two division septa (Fig 5B). Branching followed stereotypical ordered processes. First, EVs spontaneously gathered in sizable patches, reflecting positive feedbacks in EVs domain formation. These patches then fluctuated in intensity and position to eventually stabilize and promote branch emergence. Upon emergence, both growth speeds and EVs concentration increased to approach stable steady-state values within approximately 10 to 20 min (Figs 5C and S6A and S4 Movie). However, in spite of these drastic changes in both growth and vesicle concentration, the thickness of the CW did not exhibit any systematic thickening or thinning (S6A Fig). In the model, we recreated the branching process by starting with a low value of *EV*, no growth, and the reference value for CW thickness. The model faithfully reproduced the observed rapid increase and saturation in both growth speed, and EV levels with similar timescales as in experiments, as well as the near-constant CW thickness values (Fig 5C). These analyses suggest that CW synthesis and expansion increase in a balanced manner during de novo tip growth.

Conversely, we stopped growth by abruptly reducing turgor pressure. We rinsed cells with a low dose of 0.2 M sorbitol supplemented in the medium, which completely halted tip growth within a minute, for a duration of approximately 10 min. As described previously, turgor rapidly adapted and allowed growth to restart at a speed close to that before sorbitol treatment within 10 to 15 min [40,61]. Interestingly, upon sorbitol treatments, we observed a progressive delocalization of RAB11-labeled EVs domain from cell tips, which occurred slightly slower

than drops in growth speeds. These observations parallel previous reports in fission yeast [40] and further support the general hypothesis of mechanical feedback in the model. Interestingly, in contrast to de novo growth at branching sites, growth arrest upon turgor reduction was also accompanied by an increase in CW thickness of approximately 30% to 50% over 3 to 5 min, reflecting a significant transient imbalance between CW synthesis and expansion. We interpret this as a result of the slower drop of EVs concentration in comparison to CW strain rates, which presumably yield to leftover synthesis with no deformation thereby thickening the CW (Fig 5D and S5 Movie). Accordingly, when tip growth restarted, due to turgor adaptation, the EVs signal recovered its initial intensity, and CW thickness progressively decreased toward original values before the sorbitol shock. To test the model against these turgor modulations, we inputted an abrupt drop in turgor values followed by a progressive adaptation. This allowed to recapitulate growth arrest followed by growth restart, the progressive decrease of EV concentration upon turgor loss and its recovery upon growth restart, as well the dynamics of CW thickening followed by progressive thinning, though this effect was less pronounced in the model than in experiments (Fig 5E). These findings demonstrate how temporal delays between CW secretion and expansion at cell tips can transiently impact CW thickness and mechanics.

To more directly affect vesicle accumulation at cell tips, independently of turgor manipulations, we next used two independent assays to alter EV trafficking. We first depolymerized microtubules that serve as important tracks for EV polarized trafficking. We treated hyphal cells with low doses of benomyl, which caused microtubules to disappear within 2 to 4 min (S6B Fig). This led to a fraction of cells that kept on elongating at a slower rate and exhibited frequent turns in growth direction, and others that completely halted growth, and exhibited tip bulging concomitant with a progressive dispersion and loss of EVs [23,62]. Remarkably, as a consequence of growth arrest and EV reduction, the CW at the bulging tip exhibited significant thickening transiting from values of approximately 65 nm, up to approximately 100 nm in a timescale of approximately 20 min (Fig 5F and 5G and S6 Movie). We also affected vesicle accumulation using *sarA6*, a temperature-sensitive allele of *sarA* encoding the ARF GTPase governing ER exit, which results in Golgi disassembly, thereby blocking the production of post-Golgi EVs [22]. When cells were shifted to the restrictive temperature, the polar domain of RAB11 EVs completely dissipated from cell tips in a timescale of 10 to 20 min (Fig 5H and 5I and S7 Movie). Remarkably and in agreement with microtubule depolymerization experiments, this disappearance was concomitant with a growth arrest, marked bulging at cell tips, and significant CW thickening. Furthermore, growing *sarA6* for extended periods of times at restrictive temperature, yielded to large balloon-shaped tips of up to 25 μm in diameter featuring CW thicknesses reaching up to 250 nm (S6C and S6D Fig) [22]. In the model, we simulated both benomyl and *sarA6* results by reducing the source and sink terms that control the dynamic concentration of EVs. This allowed to reproduce both EV and growth reduction as well as CW thickening over similar timescales as in experiments. Therefore, in response to alterations in either growth or secretion, CW assembly appears to occur faster than expansion, ensuring that CW thicken rather than thin to safeguard cell surface integrity.

## Steady-state hyphal growth

A fascinating feature of fungal growth is the large diversity of tip growth speeds, which may span 1 to 2 orders of magnitude within a given mycelium and up to 3 to 4 orders of magnitude among different species [26]. We sought to address how turgor, CW mechanics, secretion, and expansion may be modulated to stabilize hyphal growth at different elongation speeds. For this, we exploited natural variations in tip growth speed between early germling tubes, which grow typically 4 to 5× slower than mature hyphae [23,27]. We also included in our analysis mature hyphae of a *myoVΔ*

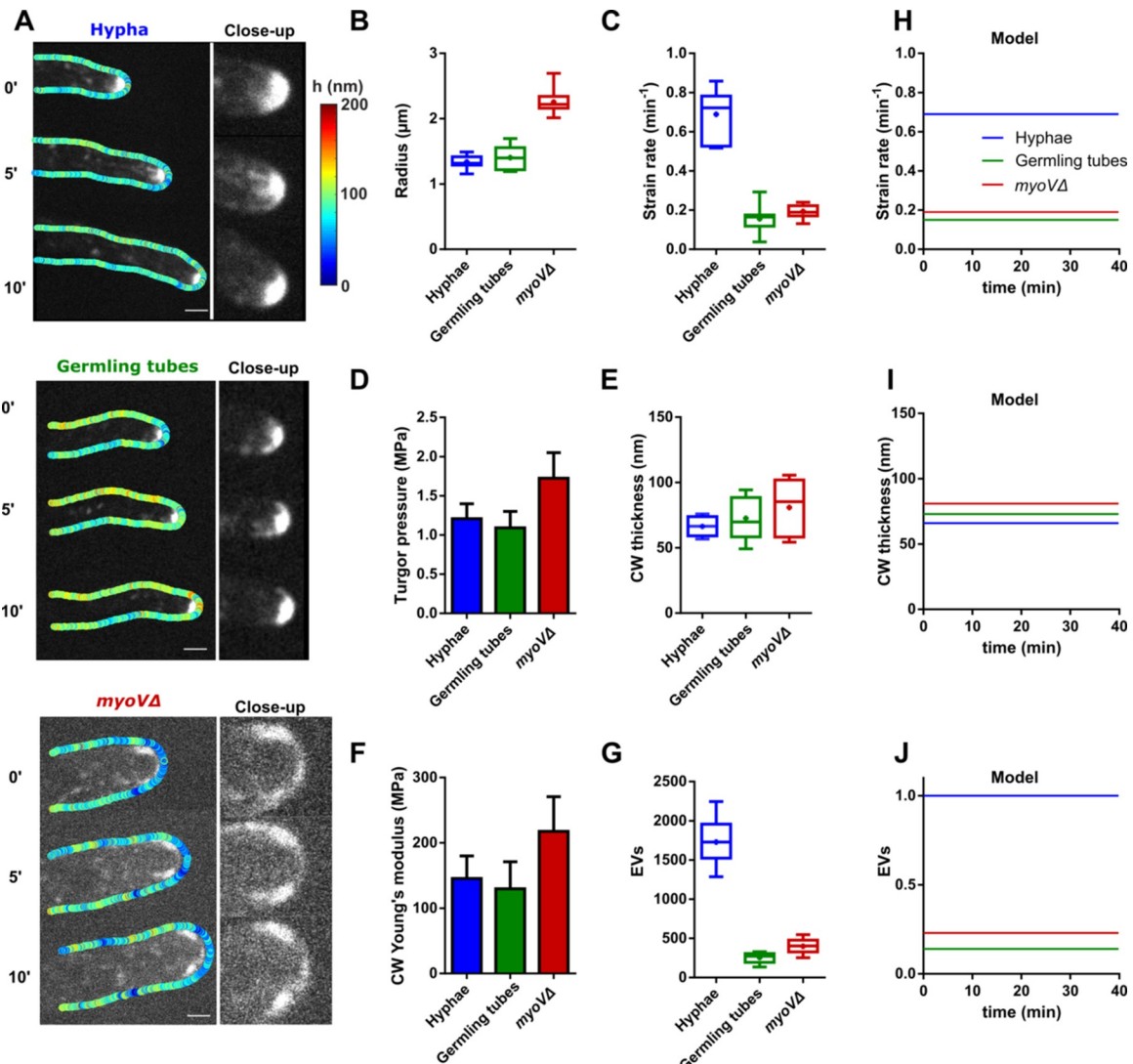

**Fig 6. CW mechanics and secretion during steady-state hyphal growth at different elongation speeds.** (**A**) Time lapses of EVs (mCherry-RAB11) and CW thickness map of mature hyphae, germling tubes, and *myoVΔ* cells, with close-up views on EVs distribution at cell tips. (**B-G**) Measured geometrical, mechanical, and biochemical parameters for mature hyphae, germling tubes, and *myoVΔ* cells (*n* > 10 cells for each conditions): Tip radius (**B**); Tip CW strain rate (**C**); Turgor pressure (**D**); Tip CW thickness (**E**); Tip CW young's modulus (**F**); and EVs tip concentration (**G**). (**H-J**) Model outputs in term of strain rates, CW thicknesses, and EVs tip concentration for the three steady-state hyphal growth at different elongation speeds. Scale bars, 2 µm. Error bars correspond to +/− SD for all panels, expect for turgor values in 6D in which they correspond to estimated errors from computing turgor by intersecting elastic strains obtained from laser ablation of the CW to that from ranges of osmotic shocks. The data underlying the graphs can be found in S1 Data. CW, cell wall; EV, exocytic vesicle.

mutant, which elongate at speeds intermediate between WT germlings and mature hyphae, but also feature a larger radius associated to more dispersed EV distribution at cell tips [15,19] (Figs 6A–6C, S7A and S7B). Interestingly, slowly elongating germling tubes exhibited similar values of turgor and CW bulk elasticity as mature hyphae, but a slightly higher thickness at cell tips. In *myoVΔ* hyphal cells, pressure, CW bulk elasticity, and thickness were all higher than in WT cells but with a relatively minor effect on tip CW elastic strain (Figs 6D–6F and S7C–S7F). These experimental results suggest that large variations in CW expansion rates and, thus, tip growth speeds may not be primarily dictated by modulations of turgor or CW mechanics.

Accordingly, quantification of mean mCherry-RAB11 levels at cell tips showed that differences in tip CW strain rates were mostly matched to differences in the apical concentration of EVs (Fig 6G). Importantly, by feeding these different mechanical and biochemical values into the model, we could identify steady-state stable solutions that recapitulated tip growth speeds, levels of secretory vesicles as well as values of tip CW thicknesses (Fig 6H–6J). Interestingly, in these steady states, the model parameter representing the mechanical coupling, $\Phi$, was found to be much higher for WT as compared to the two other conditions (S2 Table). This suggests that an adaptation of mechano-sensing and/or mechano-transduction strength to tip growth speeds may be important to stabilize different rates of steady-state growth. We conclude that modulations of tip growth speeds may mostly emerge from alterations in trafficking modules and feedbacks controlling secretory vesicles recruitment to cell tips.

## Discussion

### A new method to visualize rapid CW dynamics in tip growing cells

The structure, mechanics, and dynamics of the CW is of paramount importance for the lifestyle of all fungal species and highly relevant to host infections and antifungal agents [63,64]. Here, we adapted and validated a subresolution microscopy method initially developed for fission yeast [44], to map CW thickness in space and time in a model filamentous fungus. This adaptation rested on optimizations of growth conditions to ensure that fungal cells elongated within the focal plane and on the use of different labels suited to visualize the plasma membrane and CW surface in *A. nidulans*. As such, we anticipate that it could be easily implemented in numerous fungal species to understand how CW mechanics and dynamics are modulated during growth, infection, reproduction, or in response to antifungal chemicals that target the fungal CW. The advent of this live imaging method is multiple. First, it allows to map CW thickness along length scales of 10 to 100s of μm in live cells, which is infeasible with standard transmission electron microscopy methods (Fig 1A–1E). Second, thickness measurements allow to access local values of CW elastic moduli, from simple strain–stress assays (Fig 2B). This should be particularly valuable to address how the diversity of CW composition across fungi may impact bulk material properties. Finally, and most importantly, CW thickness dynamics provides a unique mean to monitor the balance or imbalance between CW synthesis and expansion (Fig 5). Therefore, we expect that this method could have a significant impact on our understanding of many fundamental aspects of fungal biology, physiology, and pathogenicity.

### Impact of secretory vesicles on the balance and imbalance between cell wall synthesis and deformation

Tip growth is a widespread process in fungi but also in many bacterial and plant cells, like pollen tubes and root hairs. In many cells, this process is widely accepted to be driven form the polarized recruitment of secretory vesicles that contribute to both CW assembly and remodeling. Accordingly, seminal studies based on electron microscopy of fixed plant cells, quantified CW thickness, and vesicle volumes and numbers to demonstrate a near exact mass balance between vesicle amount, CW fabrication, and deformation [65]. Other important evidence include the localization of several CW synthases in EVs [21]. In this context, the contribution of our work stands in properly computing the dynamic evolution of EVs at the apex, together with that of CW thickness and expansion, in live cells. We found that during de novo tip growth, EVs domains need to reach a threshold concentration to initiate tip growth (Fig 5B and 5C). At this level, CW material synthesis and deformation appear to be balanced, given the absence of notable thinning or thickening of the CW. As tip growth continues, this balance is maintained, as

cells expand over long length scales of 10 to 100s of μm, limiting excessive fluctuation in thickness, tip speeds, and vesicle concentration. Similarly, impairing trafficking leads to a scenario in which leftover synthesis appear to exceed deformation, generating a progressive CW thickening (Fig 5F–5I). Importantly, in all these situations, we never observed any notable systematic thinning of CWs. Therefore, a significant achievement of our method is to directly demonstrate in live cells how biochemical layers converting EVs into actual CW fabrication and deformation ensure that the CW will remain intact as tips emerge and change shape.

### A model for tip growth based on mechanical feedback

Our quantitative set of experiments allowed us to implement and systematically test a general dynamical model for tip growth. This model is based on minimal assumptions for the impact of vesicles on CW synthesis and remodeling and can account for all these results but requires a mechanical positive feedback from CW deformation rates onto vesicle concentration [35,38,39,44]. This feedback is supported experimentally by the rapid detachment of EVs when tip growth is mechanically hindered by an obstacle or slowed down by reducing turgor (Figs 5D, 5E and S5). Although multiple models have been developed in the context of tip growth [51,52], to the best of our knowledge, our work provides the first semiquantitative comparison between model outputs and measured experimental values. In addition, parameter adjustments in different situations provide interesting predictions that may guide future experiments. For instance, the model suggests that large variations in steady tip growth speeds may be accompanied by an increase in the feedback parameter $\Phi$, which couples deformation rates and vesicles accumulation. Therefore, we envision that more rapidly elongating cells may be equipped with putative CW mechanosensitive systems tailored to their rapid growth.

These considerations raise the more general question of how stress, strains, or deformation rates may be sensed in the CW, to adapt vesicle recruitment and stabilize tip growth. To date, although the regulators of membrane trafficking needed to assemble polar EV domains have been well studied in many fungal models [14,21], the mechanisms by which EVs could potentially sense and adapt to CW mechanics or dynamics remain unclear [66]. One class of plausible mechanisms may implicate conserved fungal CW mechanosensors of the WSC and MID families [67]. These act as bona fide CW mechanosensors to activate the CW integrity pathway and reinforce the CW by promoting synthesis [68]. Accordingly, previous work has shown that the *Aspergillus* WscA sensor may function to support colony growth and CW shapes, with null mutants exhibiting frequent tip bulging and lysis phenotypes in hypo-osmotic conditions [69]. Alternative mechanisms of mechanical feedbacks could emerge from stretched activated calcium channels that regulate contact sensation and tip reorientation in the fungus *C. albicans* and that have been implicated in CW integrity and polarity regulation in *A. nidulans* [58,70,71]. Finally, another possibility is that these feedbacks could be mediated at the level of upstream polarity regulators like Rho, Rac, and Cdc42 [72]. In that view, membrane addition by the fusion of EVs may result in a local dilution of these membrane-bound polarity effectors, yielding to potential homeostatic mechanisms that disperse polarity when growth occurs slower than vesicle addition [73]. Dissecting these feedbacks and their relevance to CW mechanobiology and growth is an exciting direction for fungal biology, to which dynamic methods to monitor CW properties, such as the one we developed here will be essential.

## Materials and methods

### Fungal growth conditions and medium

*Aspergillus nidulans* cells were grown from a spore solution during 16 to 20 h at 25˚C, either on WMM [74] or in complete medium (MCA) containing 1% glucose and 5 mM ammonium

tartrate, either in liquid using 8-well chambers (IBIDI GmbH, Martinsried, Germany) or in between 2% agar pads covered with a dialysis membrane and a coverslip. Spores were collected from the mycelium after approximately 15 days of growth on agar plates and resuspended in water supplemented with 0.01% of Tween. For long-term storage, spores were conserved in 20% glycerol at −80˚C. Strains used in this study were generated using standard genetic and transformation procedures [75–77] and are listed in S1 Table.

## Chemical inhibitors

The microtubule drug benomyl was used at a final concentration of 4.8 μg/mL from a 2,000X stock made in DMSO [23]. This concentration was chosen as the minimum at which microtubules were fully depolymerized, identified using a strain expressing GFP-TubA.

## Sorbitol treatments

Sorbitol treatments to measure turgor pressure were done at different concentrations ranging from 0.1 to 1.5 M sorbitol diluted in MCA medium. To arrest growth, we used a concentration of 0.2 M sorbitol, a concentration identified as the smallest to arrest growth in a large fraction of cells. These shocks were performed under the microscope while monitoring cells.

## Membrane and CW labeling

Lectins were used at 10 μg/mL for the WGA lectin, and 70 μg/mL for the ConA lectin. For membrane labeling, 1 μL of 5 μg/mL of FM4-64 diluted in PBS was added close to the hyphae of interest in liquid media, and images were taken rapidly (less than approximately 2 min) to circumvent dye endocytosis and labeling of internal compartments.

## Microfabricated channels

Microfabricated obstacles were designed and fabricated using standard methods for negative soft photolithography and polydimethylsiloxane (PDMS) prototyping [78]. In brief, a thin layer of approximately 4 to 5 μm thick of SU8 photoresist was deposited using a spin-coater (Laurell Technologies, USA) on a silicon wafer that was first washed with acetone, isopropanol, and DI water and dried. The wafer was subsequently baked and exposed to UV through a photomask containing the post designs, using a dedicated illuminator for soft lithography (Kloe, France). The wafer was post-baked and immersed in a developing chemical and finally rinsed in isopropanol and dried.

PDMS (Sylgard 184, Dow Corning, USA) was prepared at a 10:1 ratio of PDMS:curing agent, poured onto the wafer, and let to polymerize at 65˚C for 4 h. Once baked, the PDMS slabs were pierced to generate entry points and sealed with a glass coverslip using Plasma bonding (Harrick). For each experiment, spores prediluted in growth medium were injected into channels and let to germinate overnight. The channels were rinsed with fresh medium and placed under the microscope at 28˚C 1 h before imaging. Time lapses were collected at multiple positions in the channels and analyzed when hyphae grew randomly onto an obstacle.

## Microscopy

**Time lapse spinning disk confocal microscopy.** For time lapse imaging, a spore solution was deposited onto a small piece (about 2 cm × 2 cm) of dialysis membrane (300 kD, Spectra/Por 131450T), on agar medium, and grown overnight. The dialysis membrane prevents fungal hyphae from penetrating the agar. On the following day, the dialysis membrane was placed

onto a fresh agar pad supplemented with fluorescent lectins. Cells were let during 1 h for lectin to homogeneously label the CW and covered with a coverslip, and the sample was placed under the microscope, previously heated at 28˚C with an objective heater for 1 h before imaging. When grown between agar covered with the membrane and a coverlip, cell growth is confined in 2 dimensions, allowing us to perform long time lapses while keeping the mid-plane focus needed to compute CW thickness over the length of the hyphae. For temperature-sensitive alleles, we used the objective heater to raise the temperature to 37˚C, which typically occurred within 30 min. Drug addition, chemical treatment, and laser photoablation were performed in liquid media by growing cells at the bottom of specific dishes (IBIDI GmbH, Martinsried, Germany).

Images were taken using an inverted spinning-disk confocal microscope equipped with a motorized stage, automatic focus (Ti-Eclipse, Nikon, Japan), a Yokogawa CSUX1FW spinning unit, and a EM-CCD camera (ImagEM-1K, Hamamatsu Photonics, Japan), with a 2.5X additional magnifying lens, or a Prime BSI camera (Photometrics). A 100X oil-immersion objective (CFI Plan Apo DM 100X/1.4 NA, Nikon) was used. Laser ablation of the CW was performed using an iLas2 module (Gattaca, France), in the "Mosquito" mode, allowing us to perform precise CW perforation at a diffraction-limited spot with a 355-nm laser and a 60X oil-immersion objective (CFI Apochromat 60X Oil lS, 1.4 NA, Nikon). Images were recorded using the 100X objective. The microscope was piloted with Metamorph (Molecular Devices).

**Electron microscopy.** Electron microscopy was done on hyphae grown from spores overnight in rich liquid media (MCA) at the bottom of dishes. For chemical fixation, samples were fixed overnight in 2% glutaraldehyde at room temperature followed by a 1-h fixation at 4˚C in 2% osmium tetroxide in water. Cells were dehydrated using a series of ethanol solutions and embedded in epoxy resin. For HPF, samples were cryo-immobilized by HPF (EM-Pact2, Leica Microsystems). Freeze substitution was performed with an AFS2 (Leica Microsystems) in 1% osmium, 0.1% uranyl acetate, 2% water in pure acetone following protocols described previously [79]. Samples were rinsed thrice with acetone and infiltrated with gradually increasing concentrations of an Epon resin mix (Agar Scientific) and polymerized for 24 h at 60˚C. An ultramicrotome (UC6, Leica Microsystems) was used to cut 70 nm thin sections, collected on formvar/carbon-coated copper grids. The contrast was enhanced by using aqueous 4% uranyl acetate and lead citrate.

## Image analysis

**CW thickness analyses.** Based on previous work [28,44], the CW thickness was extracted using a two-color mid-slice confocal image, segmented perpendicular to the cell surface. The distance between the two signals was extracted by measuring the distance between the two Gaussian peaks. The chromatic shift was periodically and precisely calibrated using multispectral beads scanned around the field of view, and the resulting measured distance compensated. Differences in signal intensity and backgrounds were corrected using the convoluted intensity profile.

**Turgor pressure measurement.** Osmotic shocks at various sorbitol concentrations were performed on a dozen of cells imaged in rich MCA medium and imaged again within approximately 1 min of sorbitol addition. The medium osmolarity was measured using a vapor pressure osmometer (5600, Vapro, Vapor Pressure Osmometer). Cell diameters were measured manually using Image J.

To estimate turgor pressure, we used a pipeline previously established in yeast cells [53]. We first computed $\bar{c}_0$ the internal cell concentration in osmolytes, as:

$$\bar{c}_0 = \tilde{c}_0 + c_{mca} = 0.92 \ M$$

With $\tilde{c}_0$ the sorbitol concentration at which the cell shrinks as much as with photoablation, and $c_{mca}$ the molarity of the MCA rich media that we measured to be 112 mOsm.

This allowed to calculate $\bar{c}_1$, the effective concentration of the cytoplasm in the normal state, from $\frac{V_0}{V_1}$ the volume ratio before and after photoablation, and $\beta$ the inaccessible volume fraction, taken to be 0.22, with:

$$\bar{c}_1 = \bar{c}_0 \frac{\frac{V_0}{V_1} - \beta}{1 - \beta} = 0.92 * \frac{0.701 - 0.22}{1 - 0.22} = 0.517 \, M$$

This yields to an estimate of turgor, P, from the relationship:

$$P = (\bar{c}_1 - c_{mca})RT = 1.1 \text{ MPa}$$

We note that taking a value for $\beta$ of 0 yielded to higher estimates of $P$ = 1.3 MPa.

## CW Young's modulus

In order to extract the CW Young's modulus divided by pressure, the thickness of the CW was first measured, and the CW was rapidly pierced and the cell imaged again. We measured the cell radius before ($R_1$) and after ($R_0$) photoablation. We assumed that the CW is homogenous and isotropic (S2C Fig), leading to a force balance in the cylindrical part of the hyphal cell:

$$\frac{Y_{side}}{\Delta P} = \frac{R_1}{h_{side}R_1^*}$$

With $R_1^* = \frac{R_1 - R_0}{R_0}$, the elastic strain of the CW.

For the hemispherical shape of tips, force balances leads to:

$$\frac{Y_{tip}}{\Delta P} = \frac{R_{t1}}{2h_{tip}R_t^*}$$

With $R_{t1}^* = \frac{R_{t1} - R_{t0}}{R_{t0}}R_{t1}$ being the tip radius of curvature before ablation, and $R_{t0}$ after.

## Polarity domain size and concentration

The size of different polar domains was computed as the full width at mid height (FWMH) of the signal distribution along the curvature of the cell tip, using a homemade matlab script based on the FWMH function (https://www.mathworks.com/matlabcentral/fileexchange/10590-fwhm). To quantify the signal distribution, we measured the intensity of the tagged protein around cell tips, using the arc length from the pole of the cell (s) as a coordinate (s = 0 μm being the center of the tip).

The intensity of the mCherry-RAB11 at cell tips was measured by calculating the maximum intensity minus the background using a homemade Matlab script. To compensate for photobleaching, the signal was measured in time lapse of cells fixed with glutaraldehyde and used as a reference. In the particular case of branching, the EV signal was relatively low at the beginning of the process, so that measurements were done by measuring a small rectangular region intensity, minus the background, using ImageJ.

## Statistical tests

Statistical tests were done using Prism 6 software (GraphPad Software, La Jolla, CA). Data were compared by using a two-tailed Mann–Whitney test. n.s, $P > 0.05$; *, $P < 0.05$; **, $P < 0.01$; ***, $P < 0.001$; ****, $P < 0.0001$.

## Model for tip growth

As described in main text, the model for tip growth is a 1D analytical model that solves a set of coupled linear differential equations to identify parameter values. The model was solved using Mathematica. Details of all modeling procedure, results, and parameters are available in the Supporting information files (S1 and S2 Models), which can be readily viewed using Wolfram Player (https://www.wolfram.com/player/). Values of the parameters of different sets of simulations are provided in S2 Table.

The numerical data used in all figures are included in S1 Data.

## Supporting information

**S1 Fig. CW thickness measurements in different conditions, and controls for the effect of lectins and laser illumination on cell growth.** (**A**) Mid-slice confocal image of a dividing hyphae, stained with fluorescent lectins, demonstrating that lectins only decorate exposed polysaccharides at the cell surface, and not internal CWs at the septum. (**B**) Impact of lectin (WGA) and fluorescent imaging on cell growth speeds ($n > 15$ cells in each condition). (**C**) CW thickness measured with 2 different lectins: WGA ($n = 82$ cells) or ConA ($n = 89$). (**D**) CW thickness measurement using PH-GFP to label the plasma membrane ($n = 103$) or other plasma membrane marker: the FM4-64 dye ($n = 25$) or a strain expressing the membrane-associated synaptobrevin protein SSOA-GFP ($n = 49$). (**E**) CW thickness of cells grown in minimal liquid media (WMM, $n = 53$), rich liquid media (MCA, $n = 81$), or rich solid media (MCA, $n = 19$), using the WGA lectin. (**F**) Cell radii in the same conditions as in E. (**G**) CW thickness of spores, cells sides, and tips in germling tubes ($n = 24$, 11 and 11 cells). Scale bar, 2 μm. Error bars correspond to +/− SD. Results were compared by using a two-tailed Mann–Whitney test. n.s, $P > 0.05$; **, $P < 0.01$, ***, $P < 0.001$, ****, $P < 0.0001$. The data underlying the graphs can be found in S1 Data. ConA, Concanavalin A; CW, cell wall; WGA, wheat germ agglutinin; WMM, watch minimal medium.
(TIF)

**S2 Fig. Mechanical properties of the hyphal CW and cytoplasm.** (**A**) Ratio between the radius and the longitudinal shrinkage (elastic strain) during photoablation assay ($n = 10$ cells). A ratio close to 2 suggests the abscence of major anisotropies in the CW. (**B**) CW Young's divided by turgor pressure, measured along mature hyphae ($n = 7$ at least for each compartments). (**C**) Turgor pressure measurement using the percentage of plasmolyzed cells (in which the plasma membrane detaches from the CW) as a function of medium osmolarity ($n = 17$ cells at least for each molarity). Error bars correspond to +/− SD. Results were compared by using a two-tailed Mann–Whitney test. **, $P < 0.01$, ****, $P < 0.0001$. The data underlying the graphs can be found in S1 Data. CW, cell wall.
(TIF)

**S3 Fig. Comparison of CW thickness local gradients with actin-labeled endocytic collars.** (**A**) Maximum projection of cells expressing LifeAct-TdTomato overlaid with the CW thickness profile. (**B**) Measurement of the arclength distance between the two maxima corresponding to the endocytic collar, and the FWMH of the CW thickness at cell tip. (**C**) Comparison between the distance between the two maxima of LifeAct signal and the FWMH of the CW thickness. Each black line corresponds to a single cell. The data underlying the graphs can be found in S1 Data. CW, cell wall; FWMH, full width at mid height.
(TIF)

**S4 Fig. Temporal cross-correlation between thickness growth and secretion dynamics, assessment of CW advection during growth, and model outputs in the absence of**

**mechanical feedback.** (**A**) Temporal cross-correlation, as a function of different time delays between pairs of the three parameters: EVs concentration, the strain rate, and the tip CW thickness ($n = 10$ time lapses). Postive values stand for correlations; negative values are anticorrelations; and null values suggest the absence of any correlationat timescales assayed. Cross-correlations are computed as $\frac{<(h(t)-<h>)(EV(t+dt)-<EV>)>}{\sigma_h \sigma_{EV}}$. (**B**) Two representatives examples of tracking of a fiducial thickness mark where the CW is locally thicker (indicated by dotted circles) during hyphal growth, and corresponding thickness profiles plotted a function of s for different time points in the movie (color coded) and shifted to substract cell growth. The alignment of the peaks suggests that the mark is fixed with respect to the lab referential and, thus, the absence of major CW advective backward flows. (**C**) Simulation of the effect of a small postive or negative perturbation of the EV level on strain rate and CW thickness dynamics in the absence of mechanical feedback in the model. (**D**) Simulation of the effect of a small postive or negative perturbation of the EV level on strain rate and CW thickness dynamics in a model with a strain/stress-based feedback instead of a strain rate–based feedback. Scale bar, 2 μm. Error bars correspond to +/− SD. The data underlying the graphs can be found in S1 Data. CW, cell wall; EV, exocytic vesicle.
(TIF)

**S5 Fig. Evidence for mechanical feedbacks coupling growth and secretory vesicle domain stability.** (**A**) Examples of hyphae growing in PDMS microchannels that contact microfabricated posts, delineated in the fluorescence channel with yellow dotted lines. Note how EVs labeled with mCherry-RAB11 exhibit a partial dispersal followed by a domain reformation away from the site of contact. (**B**) Average dynamic evolution CW strain rate and mCherry-RAB11 levels (EVs concentration) ($n = 6$) and corresponding model outputs. The origin of time is defined as the time of contact. The CW thickness is not measured experimentally due to technical limitation in microchamber. (**C**) Dynamic evolution of the angle of polarity reorientation, computed as the angle with respect to the initial growth axis ($n = 6$). Scale bars, 2 μm. The data underlying the graphs can be found in S1 Data. CW, cell wall; EV, exocytic vesicle; PDMS, polydimethylsiloxane;
(TIF)

**S6 Fig. CW thickness local dynamic during cell branching, control for the effect of benomyl, and CW thickness in *sarA6 ts* mutants grown for long times at restrictive temperature.** (**A**) Examples of branching events. Left: CW thickness map overlaid on the EVs (mCherry-RAB11) signal. Right: Intensity profile of the EVs and CW thickness along the cell side, with s = 0 corresponding to the incipient branching site. (**B**) Depolymerization of microtubules (marked with white arrows) visualized using a strain expressing TubA-GFP upon benomyl treatment. (**C**) Brigth field image overlaid with CW thickness maps of a *sarA6 ts* cell grown 4 h at 37˚C. (**D**) Mean CW thickness of *sarA6 ts* grown at permissive temperature (28˚C) or restrictive temperature (37˚C) for 4 h. Scale bar, 2 μm. Error bars correspond to +/− SD. The data underlying the graphs can be found in S1 Data. CW, cell wall; EV, exocytic vesicle.
(TIF)

**S7 Fig. Cell growth, CW thickness, and mechanics of mature hyphae, germling tubes, and *myoVΔ* cells.** (A) Linear elongation speed of mature hyphae, germling tubes, and *myoVΔ* cells. (**B**) Surface growth speed of mature hyphae, germling tubes, and *myoVΔ* cells. (**C-D**) CW thickness (**C**) and CW Young's modulus divided by turgor pressure (Y/P) (**D**), of mature hyphae, germling tubes, and *myoVΔ* cells, at cell tips vs. cell sides. (**E**) Turgor pressure measurement from CW lateral elastic strains as a function of medium osmolarity of mature

hyphae, germling tubes, and *myoVΔ* cells. (**F**) Tip CW elastic strains of mature hyphae, germling tubes, and *myoVΔ* cells. The data underlying the graphs can be found in S1 Data. CW, cell wall.
(TIF)

**S1 Table.** *Aspergillus nidulans* **strains used in this study.**
(DOCX)

**S2 Table. Model parameters.**
(DOCX)

**S1 Models. Details of all modeling procedure, results, and parameters, which can be readily viewed using Wolfram Player (https://www.wolfram.com/player/).**
(NB)

**S2 Models. Details of all modeling procedure, results, and parameters in a .pdf file.**
(PDF)

**S1 Data. All data numbers underlying all graphs presented in main and supplemental figures.**
(XLSX)

**S1 Movie. Spatiotemporal dynamics of CW thickness of 3 growing mature** *Aspergillus nidulans* **hyphae, related to Fig 1. CW, cell wall.**
(AVI)

**S2 Movie. Spatiotemporal dynamics of CW thickness and EVs (mCherry-RAB11) of a growing mature** *Aspergillus nidulans* **hypha, related to Fig 4. CW, cell wall; EV, exocytic vesicle.**
(AVI)

**S3 Movie. Representative time lapses combining brigth field and fluorescent channels (mCherry-RAB11) of a growing hypha against a microfabricated post causing a reduction in growth, and alteration in EVs domain localisation. Related to S5 Fig. EV, exocytic vesicle.**
(AVI)

**S4 Movie. Spatiotemporal dynamics of CW thickness and EVs (mCherry-RAB11) during hyphal branching, related to Figs 5 and S5. CW, cell wall; EV, exocytic vesicle.**
(AVI)

**S5 Movie. Spatiotemporal dynamics of CW thickness and EVs (mCherry-RAB11) of a growing hypha during a turgor drop (+0.2 M sorbitol), causing a stop in growth, and a delocalization of EVs.** Note how the hypha adapts and restarts growth. Related to Fig 5. CW, cell wall; EV, exocytic vesicle.
(AVI)

**S6 Movie. Spatiotemporal dynamics of CW thickness and EVs (mCherry-RAB11) of a growing hyphae while depolymerizing microtubules with benomyl, causing a stop in growth, a delocalization of EVs and bulging of the tip.** Related to Figs 5 and S5. CW, cell wall; EV, exocytic vesicle.
(AVI)

**S7 Movie. Spatiotemporal dynamics of CW thickness and EVs (mCherry-RAB11) of a growing** *sarA6 ts* **mutant hyphae during a shift to the restrictive temperature, causing a**

**stop in growth and bulging of the tip, related to Figs 5 and S5. CW, cell wall; EV, exocytic vesicle.**
(AVI)

## Acknowledgments

We thank Sergio Fandiño for sharing the Lifeact-tdTomato strain, as well as P. Bassereau, J.M Camadro, F. Kingleschmidt, and S. Taheraly, and all members of the Minc team for discussion and technical help.

## Author Contributions

**Conceptualization:** Louis Chevalier, Arezki Boudaoud, Nicolas Minc.

**Formal analysis:** Louis Chevalier.

**Funding acquisition:** Miguel A. Peñalva, Nicolas Minc.

**Investigation:** Louis Chevalier, Nicolas Minc.

**Methodology:** Louis Chevalier, Mario Pinar, Rémi Le Borgne, Catherine Durieu, Miguel A. Peñalva, Arezki Boudaoud.

**Software:** Arezki Boudaoud.

**Supervision:** Miguel A. Peñalva, Arezki Boudaoud, Nicolas Minc.

**Writing – original draft:** Louis Chevalier, Nicolas Minc.

**Writing – review & editing:** Louis Chevalier, Miguel A. Peñalva, Arezki Boudaoud, Nicolas Minc.

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
