## [Editor Report · Decision Letter 0]

16 Jun 2022

Dear Nicolas, 

Thank you for submitting your manuscript entitled "Cell Wall Dynamics in a Filamentous Fungus" for consideration as a Research Article by PLOS Biology.

Your manuscript has now been evaluated by the PLOS Biology editorial staff as well as by an academic editor with relevant expertise and I am writing to let you know that we would like to send your submission out for external peer review.

Once your full submission is complete, your paper will undergo a series of checks in preparation for peer review. After your manuscript has passed the checks it will be sent out for review. To provide the metadata for your submission, please Login to Editorial Manager (https://www.editorialmanager.com/pbiology) within two working days, i.e. by Jun 20 2022 11:59PM.

Kind regards,

Ines

--

Ines Alvarez-Garcia, PhD

Senior Editor

PLOS Biology

---

## [Decision Letter · Decision Letter 1]

15 Aug 2022

Dear Dr Minc,

Thank you for your patience while your manuscript entitled "Cell Wall Dynamics in a Filamentous Fungus" was peer-reviewed at PLOS Biology and please accept my apologies for the delay in providing you with our decision, mainly due to the holiday period. It has now been evaluated by the PLOS Biology editors, an Academic Editor with relevant expertise, and by three independent reviewers. 

The reviews are attached below. As you will see, the reviewers find the conclusions interesting and both Reviewers 1 and 2 are very positive and support the quality of the data presented. Reviewer 3, however, is more critical and mainly raises two points that should be addressed in order for us to consider the manuscript further for publication. In the first point, the reviewer mentions that the experimental data are not enough to back up the proposed mechanistic model. In the second point, the reviewer requests that you elaborate on how the model promotes quantitative understanding of the cell wall synthesis and regulation.

In light of the reviews, we would like to invite you to revise the work to thoroughly address the reviewers' reports. Given the extent of revision needed, we cannot make a decision about publication until we have seen the revised manuscript and your response to the reviewers' comments. Your revised manuscript is likely to be sent for further evaluation by all or a subset of the reviewers.

**IMPORTANT - SUBMITTING YOUR REVISION**

3. Resubmission Checklist

a) *PLOS Data Policy*

b) *Published Peer Review*

d) *Blurb*

Please also provide a blurb which (if accepted) will be included in our weekly and monthly Electronic Table of Contents, sent out to readers of PLOS Biology, and may be used to promote your article in social media. The blurb should be about 30-40 words long and is subject to editorial changes. It should, without exaggeration, entice people to read your manuscript. It should not be redundant with the title and should not contain acronyms or abbreviations. For examples, view our author guidelines: https://journals.plos.org/plosbiology/s/revising-your-manuscript#loc-blurb

Sincerely,

Ines

--

Ines Alvarez-Garcia, PhD

Senior Editor

PLOS Biology

Reviewers' comments

Rev. 1:

This article, by Dr. Minc, Boudaoud, and coworkers, presents a beautiful study of cell wall dynamics. The method and vision of this study is very unique, from a physical/biophysical perspective. The manuscript mapped out the cell wall thickness, stiffness of the hyphae of living A. nidulans, which provides insight into the mechanical properties associated with hyphal tip growth. It is then followed by a discussion of the cell wall seceration by monitoring the vesicle accumulation.

Though from a physical view, the manuscript is well written and should be fully accessible to any reader of PLOS Biology. I enjoyed reading the manuscript, especially for the well-organized discussions where the methodology merit, the link of secretory vesicles and cell wall balance, and cell wall mechanical feedbacks, are discussed in great details. 

The figures are highly attractive and can efficiently support the major statements of this manuscript. 

Considering all the merits, I suggestion publication on PLOS Biology. I believe it will be an important addition to the literature and will attract readers from many research directions, in particular, cell wall structural biology and microbiology. 

There is a minor question regarding the section of "A spatial gradient of cell wall stiffness is associated to hyphal polar growth," the authors mentioned that CW thickness polarity could be inverted in a time-course as short as ~few min. It will be very helpful if the authors could clarify here what is the time resolution regarding the method. 

Rev. 2:

Fungal tip growth is a subject of considerable importance. Fungi are very serious human, veterinary and plant pathogens, killing roughly 1.5 million persons each year and about 10% of harvested crops are spoiled by fungi. On the other hand fungi produce billions of dollars of industrial enzymes, small molecules such as citric acid and secondary metabolite drugs such as the statins. A great deal of progress in understanding, at a molecular level, how fungi grow has been made in recent years. We know the identities many of the molecules that are required for tip growth and we have a rough outline of the mechanisms of vesicular trafficking, but many key questions remain including the question of how vesicular trafficking is regulated.

The authors employed a method of measuring the thickness of the cell wall based on a method developed in Schizosaccharomyces pombe. It uses fluorescently tagged lectins (Concanavalin A and Wheat Germ Agglutinin) to label the outside of the cell wall and GFP-tagged Pleckstrin Homology domain to label the plasma membrane. The authors used spinning disk confocal microscopy with gaussian fitting to determine wall thickness with resolution below that of the traditional (Abbé) limit of resolution. Essentially, the distance between the gaussian peaks of the GFP-Pleckstrin Homology domain and the lectin is taken as the wall thickness. The values obtained are similar to those obtained with electron microscopy after chemical fixation or rapid freezing and the technique appears to give valid data. The technique is powerful because it allows one to measure wall thickness in living material and, in the same cells at the same time, look at dynamics of important components of the tip growth apparatus such as secretory vesicles. 

Perhaps surprisingly, thickness of the cell wall did not differ a great deal between the hyphal tip and the hyphal wall. Thickness did not change very much as a function of speed, and although tip cell wall thickness was generally slightly less than the thickness of the hyphal wall, this was not always the case, which suggests that cell wall thickness gradients do not establish polarity in tip growth. By perturbing tip growth in a number of ways (depolymerizing microtubules with benomyl, using a MyoV deletant, laser ablation, altering osmolarity), the authors were able to calculate or estimate a number of values (e.g. Young's modulus), draw a number of conclusions, and develop a mathematical model for tip growth. There is no point in reiterating all of the data and conclusions in the manuscript, but some of the main conclusions are as follows. 

The position of secretory vesicles is a good indicator of cell wall remodeling and speed of tip growth, but turgor and cell wall mechanics do not correlate well with tip growth speed. Relatedly, there is a balance of secretory vesicle recruitment and tip expansion. 

During lateral branch growth, there appears to be a positive feedback mechanism that leads to an accumulation of secretory vesicles. 

There is a spatial gradient of cell wall stiffness with cell tip walls being less stiff than lateral walls (presumably reflecting cross-linking of cell wall components?). 

The model requires a mechanical feedback loop and a number of findings indicate a positive feedback mechanism that coordinates tip growth with vesicle trafficking. For example, impairing vesicle trafficking does not result in a thinner wall, as one might intuitively assume, but, rather a thicker wall (i.e. tip growth is slowed to a greater extent than vesicle trafficking). There are plausible mechanisms that might employ conserved fungal mechanosensors.

This manuscript combines biology, sophisticated microscopy and mathematical modeling to give fresh insights into an important phenomenon. I feel that I am quite competent to evaluate the microscopy and biology, and I believe they are sound and worth of publication in a high profile journal. While the mathematical model leads to biologically plausible conclusions, I don't feel that I am capable of reviewing the mathematics in this manuscript. Indeed, one problem with multidisciplinary approaches is that it is difficult for individuals to evaluate them. Teams are really necessary. Assuming that other reviewers are happy with the mathematics, I am happy to recommend publication in PLoS Biology.

Rev. 3:

This work studies the dynamics of cell wall secretion, remodeling, and deformation of fungus hyphae. The authors developed a live-imaging technique to monitor the dynamics of hyphal cell wall (CW) thickness in Aspergillus nidulans by labeling the plasma membrane and the polysaccharides outside the CW. Combining measurement of CW thickness, turgor pressure, and the shape changes after photoablating the CW with a linear elastic model uncovered a small gradient of CW thickness and a larger gradient of CW stiffness from the hyphal tip to the side, with the tip on average being ~13% thinner but ~1/3 of Young's Modulus compared to the side. Fluorescent labeling of exocytic vesicles showed a similar spatial distribution compared to the CW thickness profile near the tip. Time-lapse imaging revealed low temporal fluctuation (~10-15%) in growth speed, CW thickness and vesicle amount near the tip. Based on these observations, the authors developed a model (ODEs) that uses posited mechanical feedback to connect growth, CW thickness, CW remodeling, and secretory vesicle production to account for steady state hyphal growth. Simulation also agreed semi-quantitatively with four test cases of abrupt changes in growth or secretion (lateral branching, reduced turgor pressure, and altered vesicle trafficking). The authors also fitted the model to the steady state growth of early germling tubes and the mature hyphae of a myoV deletion mutant and found a substantial difference in the coefficient of mechanical feedback.

Major concerns

1. The experimental data are not enough to back up the mechanistic model that integrates growth, CW thickness, vesicles production, and mechanical feedback. Live-imaging shows a relatively stable growth speed, CW thickness and vesicle amount during steady state growth; mechanical modeling suggests lower stiffness at the tip versus the side. However, the paper does not provide direct evidence of mechanics playing a role in regulating the cell wall synthesis process. There is no data backing up the mechanical feedback proposed in the model; it is also questionable that the mechanical feedback is based on strain rate-dependent vesicle production/transport, as suggested by the model.

2. The authors need to clarify the contribution of the work. Experimentally, they developed a live-imaging technique to track cell wall thickness dynamics during growth, measured the turgor pressure, and by mechanics calculation, showed a gradient in stiffness along the hypha. The results are mostly phenomenological. The mechanistic aspect of this work lies mostly in the model and fitting the model to test cases. However, the model itself does not explain its superiority to the existing models. The authors need to elaborate on how the model promotes quantitative understanding of the cell wall synthesis and regulation.

Minor concerns

1. The authors need to clarify the conclusions of the section "Spatial patterns of secretory vesicle accumulation and cell wall mechanics", which says "CW remodeling at cell tips may be best represented by the global distribution of RAB11 EVs" and "these analyses directly highlight in living cells the spatial relationships between exocytic carrier distribution at cell tips and local modulations in CW synthesis and mechanics". What does it mean and why is it important to "represent" CW remodeling? And why is the "spatial relationship" important? Is the localization of the exocytic vesicles part of the paper's discovery? Is purpose of this section mainly to justify the involvement of exocytic vesicles in the model to be preposed later?

2. In Fig. 4C, the relative standard deviation of the thickness (h) and the EV amount are above the 10-15% the authors claimed in the text.

3. In the section "Dynamic co-evolution of CW thickness, tip growth and exocytosis during hyphal tip shape changes", it says "These patches then fluctuated in intensity and position, reflecting negative feedbacks". Why does this imply negative feedbacks?

4. In the simulation of the 2 cases where vesicle accumulation is affected (Fig. 5FGHI), both EV production and degradation need to be reduced (and the degradation coefficient has to drop more than the production coefficient, as indicated in the Supplementary information) in the model to reproduce cell wall thickening. Why would microtubule depolarization/Golgi disassembly affect EV degradation more than production?

---

## [Editor Report · Decision Letter 2]

13 Dec 2022

Dear Dr Minc,

Thank you for your patience while we considered your revised manuscript entitled "Cell Wall Dynamics in a Filamentous Fungus" for publication as a Research Article at PLOS Biology. This revised version of your manuscript has been evaluated by the PLOS Biology editors and the Academic Editor.

Based on the reviews, we are likely to accept this manuscript for publication, provided you satisfactorily address the data and other policy-related requests stated below.

In addition, we would like you to consider a suggestion to improve the title:

"Cell wall dynamics stabilize fungal tip growth and promote morphogenetic plasticity in a filamentous fungus"

or

"Aspergillus cell wall dynamics stabilize fungal tip growth and promote morphogenetic plasticity"

We expect to receive your revised manuscript within two weeks. 

*Published Peer Review History*

*Press*

Sincerely,

Ines

--

Ines Alvarez-Garcia, PhD

Senior Editor

PLOS Biology

DATA POLICY:

Many thanks for submitting a data file containing the data underlying all the graphs shown in the figures. I have checked the file and I have the followin queries:

- Please add to the file the data underlying Fig. 1C and Fig. S6A or let me know where can we find this data.

- For Fig. 6D and F please add all replicates and not only the mean/average values.

- Please also mention in the figure legends where the underlying data can be found - for example, you can add "The data underlying the graphs can be found in Data_S1"

---

## [Editor Report · Decision Letter 3]

22 Dec 2022

Dear Dr Minc,

Thank you for the submission of your revised Research Article entitled "Cell Wall Dynamics Stabilise Tip Growth in a Filamentous Fungus" for publication in PLOS Biology. On behalf of my colleagues and the Academic Editor, Aaron Mitchell, I am delighted to say that we can in principle accept your manuscript for publication, provided you address any remaining formatting and reporting issues. These will be detailed in an email you should receive within 2-3 business days from our colleagues in the journal operations team; no action is required from you until then. Please note that we will not be able to formally accept your manuscript and schedule it for publication until you have completed any requested changes.

PRESS

Sincerely, 

Ines

--

Ines Alvarez-Garcia, PhD

Senior Editor

PLOS Biology
